# Soil microbial community responses to short-term nitrogen addition in China's Horqin Sandy Land

Niu Yayi[1,2,3], Duan Yulong[1,3]*, Li Yuqiang[1,2,3], Wang Xuyang[1,3], Chen Yun[1,2,3], Wang Lilong[1,3]

1 Northwest Institute of Eco-Environment and Resources, Chinese Academy of Sciences, Lanzhou, China, 2 University of Chinese Academy of Sciences, Beijing, China, 3 Naiman Desertification Research Station, Northwest Institute of Eco-Environment and Resources, Chinese Academy of Sciences, Tongliao, China

* duanyulong@nieer.ac.cn

**Data Availability Statement:** All sequencing data associated with this study have been deposited at the NCBI Sequence Read Archive (https://www.ncbi.nlm.nih.gov/sra) under project accession

## Abstract

Anthropogenic nitrogen (N) addition has increased soil nutrient availability, thereby affecting ecosystem processes and functions in N-limited ecosystems. Long-term N addition decreases plant biodiversity, but the effects of short-term N addition on soil microbial community is poorly understood. The present study examined the impacts of short-term N addition ($NH_4NO_3$) on these factors in a sandy grassland and semi-fixed sandy land in the Horqin Sandy Land. We measured the responses of soil microbial biomass C and N; on soil β-1,4-glucosidase (BG) and β-1,4-N-acetylglucosaminidase (NAG) activity; and soil microflora characteristics to N additions gradient with 0 (control), 5 (N5), 10 (N10), and 15 (N15) g N $m^{-2}$ $yr^{-1}$. The soil microbial biomass indices, NAG activity, and soil microflora characteristics did not differ significantly among the N levels, and there was no difference at the two sites. The competition for N between plants and soil microbes was not eliminated by short-term N addition due to the low soil nutrient and moisture contents, and the relationships among the original soil microbes did not change. However, N addition increased BG activity in the N5 and N10 additions in the sandy grassland, and in the N5, N10, and N15 additions in the semi-fixed sandy land. This may be due to increased accumulation and fixation of plant litter into soils in response to N addition, leading to increased microbial demand for a C source and increased soil BG activity. Future research should explore the relationships between soil microbial community and N addition at the two sites.

## Introduction

Nitrogen (N) is the major growth-limiting elements for plant growth in most terrestrial ecosystems, especially in arid and semi-arid ecosystems [1,2]. Changing N availability is therefore an important component of the functions of terrestrial ecosystems, particularly under global climate change scenarios [3,4]. During the 20th century, humans have more than doubled the amount of N added to the biosphere [5]. Anthropogenic N addition in N-limited ecosystems is

number PRJNA615072. Other relevant data are within the paper and its Supporting Information files.

**Funding:** Li Yuqiang: National Key Research and Development Program of China (2017YFA0604803 and 2016YFC0500901), the National Natural Science Foundation of China (grants 31971466 and 31400392), the One Hundred Person Project of the Chinese Academy of Sciences (Y551821); Duan Yulong: the "Light of West China" Program of the Chinese Academy of Sciences (E0290501).

**Competing interests:** The authors have declared that no competing interests exist.

a primary component of global change, as it can influence the biogeochemical coupling of the soil carbon (C) and N cycles by altering organic matter decomposition [6], and it can profoundly alter soil microbial communities and their enzyme activities [7,8].

Arid and semi-arid ecosystems cover one-third of the world's land surface and account for approximately 15% of the global soil organic carbon pool; they therefore play an important part in maintaining the world's ecosystem functions [9,10]. The Horqin Sandy Land is the largest sandy land in China, and comprises a severely desertified area in China's agro-pastoral ecotone, which has undergone tremendous changes in climate, land use, and anthropogenic N addition [11]. Sandy grassland and semi-fixed sandy land ecosystems are sensitive to increased atmospheric N deposition [12]. The availability of N is an important driver of soil enzyme activity, microbes, and soil microflora characteristics in this area [4,13].

Soil microbes are a highly sensitive and active component of terrestrial ecosystems, as they respond quickly to environmental changes, such as short-term N enrichment of terrestrial ecosystems, by driving changes in biomass activity and nutrient cycling, as well as changes in the soil microbial biomass, quantity, community structure, and diversity, as well as in soil enzyme activity [14–16].

Soil microbial biomass refers to the volume of soil less than 5000 $\mu m^3$ of total living organisms excluding plant bodies, and which is the most active component of soil organic matter and the most active factor in the soil [17]. Soil microbial biomass is the driving force for the transformation and cycling of soil organic matter and soil nutrients; it is also a reserve for soil nutrients and an important source of nutrients that are available for plant growth, and can therefore be used as an important indicator of soil fertility [18–20]. Previous studies have shown that long-term N addition reduces soil microbial biomass [21–23]. Liu et al. (2010) [15] and Li et al. (2010) [24] showed that long-term N addition decreases soil microbial biomass carbon (SMBC) and nitrogen (SMBN) in temperate steppe and sandy grassland ecosystems in semi-arid areas. However, studies of the effects of short-term N addition on soil microbial biomass in arid and semi-arid ecosystems show considerable disagreement, with researchers reporting increases [25], decreases [24,26], and no influence [27]. Since soil microbial biomass has such an important effect on nutrient transformations and flows between the soil and plants, it is particularly important to learn their response to N addition to improve our understanding of the mechanisms that underlie nutrient cycling.

Soil enzymes, which are mainly released by soil microbes, play a key role in the decomposition of soil organic matter [28,29]. Soil β-1, 4-glucosidase (BG) hydrolyzes disaccharides and trisaccharides from cellulose to produce smaller molecules, such as glucose, and has been used to characterize C cycles in the soil [30]. Soil β-1, 4-N-acetylglucosaminidase (NAG) participates in the N cycle and is secreted by microbes to hydrolyze chitin and peptidoglycan to produce glucosamine [31]. Soil dehydrogenase (DHA) is mainly found in living cells, and can be used to characterize the overall activity of microbes [32]. Changes in the activities of these soil enzymes can directly reflect the intensity and speed of soil nutrient release by decomposition of organic matter [33].

Numerous studies have investigated the responses of microbial communities to short-term and long-term nitrogen addition. However, the effects of nitrogen addition on soil microbial communities remain controversial, such as in terms of magnitude and regarding the direction. The inconsistent results among studies may be due to the heterogeneity of biome types, N application rates and types, and even experimental durations [34]. Studies in both forests [35] and grasslands [36,37] have reported that N addition increased bacterial biomass and decreased fungal biomass in grassland soil based on a long-term experiment, whereas Zheng et al. (2015) [38] found that N enrichment had no effect on fungal biomass but significantly decreased bacterial biomass in subtropical forests over 8 years. However, no significant effects

of N addition on the bacterial or fungal biomass were detected in the short-term experiment [37,39–41].

The effect of long-term N addition on soil microbial biomass and soil enzyme activity in farmland and forest ecosystems is reasonably well understood, but the feedbacks among soil microbial biomass, soil enzyme activity, and soil microflora characteristics during the response to short-term N addition in sandy grassland and semi-fixed sandy land ecosystems requires further exploration. According to studied of Bradley et al. (2006) [37] and Wan et al. (2015) [39] have shown that short-term N additions to sandy grassland and subtropical coniferous and broadleaf forest plantations. We hypothesized that short-term N additions may do not change the biomass and structure of the soil microbial community. In the present study, we obtained data to provide a clearer picture of these feedbacks.

## Materials and methods

### Site description and experimental design

The two sampling sites were established in a sandy grassland and a semi-fixed sandy land near the Naiman Desertification Research Station of the Chinese Academy of Sciences (42˚55′N, 120˚42′E), in a semi-arid region of China's Horqin Sandy Land. The distance between the two sampling sites was about 1.5 km. The terrain at the study site is flat and open, with an elevation of 377 m asl. The region has a continental semi-arid monsoon temperate climate, with an annual mean temperature of 6.8˚C, with mean monthly temperatures ranging from -9.63˚C in January to 24.58˚C in July, and with an annual mean precipitation of 360 mm, 70% of which occurs during the period from May to September. The soils of the two sampling sites were chestnut soils (Chinese soil classification). Table 1 summarizes the physical and chemical properties, initial values of soil microbial indices, and enzyme activity of the topsoil (to a depth of 20 cm) at both sampling sites. The dominant native plant species of the sandy grassland were *Messerschmidia sibirica*, *Setaria viridis*, and *Eragrostis pilosa*, and those of the semi-fixed

**Table 1. The physical and chemical properties, initial values of soil microbial indices, and enzyme activity of the topsoil (to a depth of 20 cm) at the sandy grassland and semi-fixed sandy land sites.**

| Parameter | Sandy grassland | Semi-fixed sandy land |
|---|---|---|
| SOC (g kg$^{-1}$) | 1.67±0.001a | 3.04±0.001b |
| TN (g kg$^{-1}$) | 0.12±0.009a | 0.32±0.014b |
| TP (g kg$^{-1}$) | 0.20±0.013a | 0.19±0.005a |
| pH | 8.15±0.027a | 8.50±0.150b |
| EC (μS cm$^{-1}$) | 16.76±0.517a | 26.75±0.680b |
| SMBC (mg kg$^{-1}$) | 32.24±2.600a | 26.05±2.706a |
| SMBN (mg kg$^{-1}$) | 4.80±0.766a | 5.86±0.734a |
| BG (U g$^{-1}$) | 5.29±0.353a | 16.32±1.165b |
| NAG (U g$^{-1}$) | 0.89±0.143a | 2.30±0.323b |
| DHA (U g$^{-1}$) | n.d. | n.d. |

Note: Values of a parameter followed by different letters differ significantly between the two sampling sites (One-way ANOVA followed by LSD test, $P < 0.05$). SOC, soil organic C; TN, total nitrogen; TP, total phosphorus; EC, electric conductivity; SMBC, soil microbial biomass carbon; SMBN, soil microbial biomass nitrogen; BG, soil β-1,4-glucosidase activity; NAG, soil β-1,4-N-acetylglucosaminidase activity; DHA, soil dehydrogenase activity; n.d., not detected.

Values represent means ± SD ($n$ = 24).

sandy land were *Caragana microphylla*, *Setaria viridis* and *Echinops gmelinii*. The vegetation cover was 60 and 30%, respectively.

We established 24 plots, each 1 m × 1 m, in May 2019. The treatments were a control and nitrogen addition at 5 g N m$^{-2}$ yr$^{-1}$ (N5), 10 g N m$^{-2}$ yr$^{-1}$ (N10), and 15 g N m$^{-2}$ yr$^{-1}$ (N15), are these values based on current atmospheric deposition levels (0.50 g N m$^{-2}$ yr$^{-1}$) and the predicted levels in 10 (N5), 20 (N10) and 30 (N15) years [42]. The blocks were separated by a 2.0-m-wide buffer strip, and the plots within each block were separated by a 1.0-m-wide buffer strip to minimize disturbance from neighboring plots. Nitrogen (NH$_4$NO$_3$) addition was applied once, before it rained, in mid-May 2019.

We used a 2.5-cm-diameter auger to collect topsoil samples (to a depth of 20 cm) on 15 May 2019, early in the growing season, and on 15 August 2019, at the peak of the growing season from minimally disturbed natural soils. At each site, we collected topsoil at five random locations within each plot (1 m ×1 m) and homogenized them to provide a single composite soil sample, which we packed in sterilized polyethylene bags and transported to the lab in coolers portable car refrigerators as quickly as possible. All visible roots, residues, and stones were removed by sieving (with a 2-mm square-aperture mesh). Every sample was divided into two equal subsamples. One was stored at 4˚C to determine the soil properties, and the other was stored at -80˚C until to DNA extraction.

## Measurement of microbial biomass

We used a fumigation-extraction method to measure SMBC and SMBN [43]. In summary, three fresh 50-g soil samples were placed in separate 100-mL beakers, and were then incubated in the dark for 7 days at 25˚C and a relative humidity of 70%. One soil sample was used as the control, and another was fumigated for 24 h with ethanol-free CHCl$_3$. The last soil sample was used to measure the soil moisture content. The control and the fumigated soil samples were transferred into 250-mL Erlenmeyer flasks, then 100 mL of 0.5 M K$_2$SO$_4$ was added, and the solution was shaken for 30 min at 25˚C to obtain soil extracts. Extracts were filtered through 0.45-μm cellulose ilters and stored at -20˚C until analysis. The SMBC and SMBN contents were measured using an Elementar Vario TOC (Elementar, Langenselbold, Germany). SMBC and SMBN were calculated from the difference between the extractable C and N contents in the fumigated and control samples using conversion factors: kEC for C and kEN for N were both equal to 0.45.

## Enzyme activity

The enzyme activities of the soil BG and NAG were quantified using commercial enzyme kits following the manufacturer's protocol (BG Assay kit and NAG Assay kit; Solarbio, Beijing, China). Briefly, BG decomposes p-nitrobenzene-β-D-glucopyranoside to form p-nitrophenol, and NAG decomposes p-nitrobenzene β-N-acetylglucosamine to also form p-nitrophenol, which has a maximum absorption peak at 400 nm. We used a UV-VIS spectrophotometer (UV-1800, Mapada Instruments Co., Shanghai, China) to measure the absorbance. BG and NAG activities were calculated by measuring the rate of increase in absorbance. DHA activity was also measured using a commercial enzyme kit (the DHA Assay kit, Solarbio). 2, 3, 5-triphenyl tetrazolium chloride is reduced to triphenyl formazone after receiving hydrogen during cell respiration. Triphenyl formazone is red and has a maximum absorption peak at a wavelength of 485 nm, and its absorbance was also measured by UV-VIS spectrophotometry to obtain the DHA activity.

## DNA extraction

From each sample, total DNA was extracted from 0.5 g of soil using the PowerSoil kit (Omega Laboratories Inc., Mogadore, OH, USA) according to the manufacturer's instructions. The

integrity of the DNA was determined by electrophoresis in 1.0% agarose gels, and the purity and concentration of the DNA were measured spectrophotometrically with a NanoDrop ND5000 (Thermo Fisher Scientific Inc., USA).

## Quantitative real-time polymerase chain reaction

The polymerase chain reaction (PCR) was performed using a Line-Gene 9600 Plus Cycler (Thermo Fisher Scientific Inc.). The hyper-variable 444 bp V3 to V4 region of the bacterial 16S rRNA was amplified for each sample using two primers (338F, 5′–ACTCCTACGGGAGGCAG CAG–3′; 806R, 5′–GGACTACHVGGGTWTCTAAT–3′) [44]. Similarly, the 317-bp ITS1 region of the fungal ITS rRNA was amplified for each sample using two primers (ITS1F, 5′–CTTG GTCATTTAGAGGAAGTAA–3′; ITS2R, 5′–GCTGCGTTCTTCATCGATGC–3′) [45].

To estimate bacterial and fungal small-subunit rRNA gene abundances, we generated standard curves using a 10-fold serial dilution with a plasmid containing a full-length copy of either the *Escherichia coli* 16S rRNA gene or the ITS rRNA gene. Quantitative PCR (*q*PCR) was performed with 25 mg of the sample mixed with 12.5 mL of ChamQ SYBR Color qPCR Master Mix (2X) (Vazyme Biotech Co., Ltd, Nanjing, China), 0.5 mL solutions (10 mM) of each forward and reverse primer, and 9.5 mL of sterile, double-distilled H$_2$O. Standard and environmental DNA samples were added at 2.0 mL per reaction. The reaction was carried out on a Line-Gene 9600 Plus Cycler (Thermo Fisher Scientific Inc.). The cycling program was an initial denaturation at 95˚C for 3 min, followed by 40 cycles of 94˚C for 30 s, 53˚C for 30 s, and 72˚C for 45 s, with a final extension at 72˚C for 5 min. Melting curve and gel electrophoresis analyses were performed to confirm that the amplified products were of the appropriate size. Bacterial and fungal gene copy numbers were generated using a regression equation for each assay that related the cycle threshold (*Ct*) value to the known number of copies in the standards. All of the qPCR reactions were run in triplicate for each soil sample. The average bacterial PCR efficiency was 92.22% with an $R^2$ of the standard curves of 0.9991, and the fungal PCR efficiency was 91.99% with an $R^2$ of the standard curves of 0.9995.

## PCR amplification and illumina MiSeq sequencing

PCR was carried out in triplicate in a 20-μL reaction volume that contained 4 μL of 5-fold reaction buffer, 4 μL of dNTPs (2.5 mM), 0.8 μL of each primer (5 μM), 1 μL of template DNA (ca. 10 ng), and 0.4 μL of Pfu DNA Polymerase (TransStart-FastPfu DNA Polymerase, TransGen Biotech, Beijing, China), with double-distilled H$_2$O to bring the solution to the final volume. The PCR program included an initial denaturation at 95˚C for 3 min; 35 cycles at 94˚C for 30 s, annealing at 55˚C for 30 s, and an extension at 70˚C for 45 s; and a final extension at 72˚C for 10 min. PCR was performed using an ABI GeneAmp 9700 Cycler (Thermo Fisher Scientific Inc.). We have checked for inhibition test to make sure the accuracy of qPCR data.

Different barcode sequences were added at the 5′ end of the forward primer to separate corresponding reads from the data pool that was generated in a single sequencing run. The amplicons were extracted by electrophoresis in 2.0% agarose gels, purified by using a Gel Extraction Kit (Axygen Co., Hangzhou, China) according to the manufacturer's instructions, and quantified using a QuantiFluor-ST Fluorimeter (Promega, Fitchburg, WI, USA). The purified amplicons were pooled in an equimolar and paired-end sequence (2×300) on an Illumina MiSeq PE300 Sequencer (Majorbio Co. Ltd., Shanghai, China) according to the manufacturer's standard protocols.

**Statistical analysis, processing, and analysis of the sequencing data.** We tested for differences in the soil properties, soil microbial biomass indices, and soil enzyme activity between the sandy grassland and semi-fixed sandy land with different N addition levels using one-way

analysis of variance (one-way ANOVA). Site type and N addition were used as treatment factors to conduct two-factor ANOVA for soil microbial indicators and enzyme activity. The data were tested to confirm normality and homogeneity of variance (Levene's test) prior to ANOVA. When the ANOVA results were significant, we used the least-significant-difference test to identify significant differences between pairs of values, with significance at $P < 0.05$. The analyses were performed using version 19.0 of the SPSS software (https://www.ibm.com/analytics/spss-statistics-software).

Raw FASTQ files were de-multiplexed and quality-filtered using version 0.35 of the Trimmomatic software (http://www.usadellab.org/cms/?page=trimmomatic) with the following criteria: (i) The 300-bp reads were truncated at any site that obtained an average quality score less than 20 over a 50-bp sliding window, and truncated reads shorter than 50 bp were discarded. (ii) We removed the extracted matching barcodes, and any two-nucleotide mismatches in the primer matching and reads that contained ambiguous characters. (iii) Only overlapping sequences longer than 10 bp were assembled according to their overlapping sequence. Reads that could not be assembled were discarded.

Quality sequences were aligned in accordance with the SILVA alignment database (https://www.arb-silva.de/) [46] and clustered into operational taxonomic units (OTUs) using version 7.1 of the USEARCH software (https://www.drive5.com/usearch/). OTUs with a 97% or better similarity level were used for the rarefaction curve, and we calculated the α-diversity indices, including the ACE, Chao, Shannon, and Simpson diversity indices, and performed coverage analysis using version 1.30.2 of the mothur software (https://www.mothur.org/) [47]. Taxonomic assignments of the OTUs with at least 97% similarity were performed using mothur in accordance with the SILVA (132) or Unite (8.0) databases with a 70% confidence interval. For taxonomic analysis, we used the SILVA database and the Unite database (http://unite.ut.ee/index.php) for bacteria and fungi, respectively. For β-diversity analysis, we performed principal-components analysis (PCA) and generated a hierarchical heatmap using version 2.5–6 the vegan package (https://cran.r-project.org/web/packages/vegan/index.html) for version 3.2.0 of the R statistical software (https://www.r-project.org/).

**Data deposition.** All sequencing data associated with this study have been deposited at the NCBI Sequence Read Archive (https://www.ncbi.nlm.nih.gov/sra) under project accession number PRJNA615072.

## Results

### Changes in microbial biomass indices

In the sandy grassland, the soil biomass microbial indices (SMBC, SMBN, and SMBC/SMBN) did not differ significantly among N additions ($P > 0.05$, Table 2). All N addition levels were significantly decreased SMBC, and control level was significantly decreased SMBN compared with the background ($P < 0.05$, Table 2). In the semi-fixed sandy land, N addition significantly decreased SMBC and SMBN compared with the control ($P < 0.05$, Table 2), but with no significant differences among N addition levels ($P > 0.05$). SMBC/SMBN did not differ significantly from the control at any N addition level. In compared with background, control level was significantly increased SMBC, and all N addition levels were significantly increased SMBC/SMBN ($P < 0.05$, Table 2). We also compared the soil microbial biomass indices in a given treatment between the sampling sites (Table 2). In the control, SMBC and SMBN were significantly higher in the semi-fixed sandy land ($P < 0.05$), but their ratio did not differ significantly. However, the SMBC, SMBN, and SMBC/SMBN did not differ significantly between the two sampling sites in any N treatment ($P > 0.05$). In the background, the SMBC/SMBN of sandy grassland were significantly higher in the semi-fixed sandy land ($P < 0.05$).

**Table 2. The soil microbial biomass indices (soil microbial biomass carbon (SMBC), soil microbial biomass nitrogen (SMBN), and SMBC/SMBN ratio) and soil enzyme activities (β-1,4-glucosidase (BG), soil N-acetyl-β-D-glucosidase (NAG), and soil dehydrogenase activity (DHA)) in the topsoil (to a depth of 20 cm) in the sandy grassland and semi-fixed sandy land.**

| Parameter | Sandy grassland | | | | | Semi-fixed sandy land | | | | |
|---|---|---|---|---|---|---|---|---|---|---|
| | Background | Control | N5 | N10 | N15 | Background | Control | N5 | N10 | N15 |
| SMBC (mg kg$^{-1}$) | 32.24 ±2.60Ab | 17.16 ±2.59Aa | 16.32 ±4.37Aa | 16.81 ±4.45Aa | 13.48 ±1.97Aa | 26.05 ±2.71Aa | 62.99 ±14.02Bb | 25.50±1.98Aa | 20.36±6.87Aa | 23.57 ±3.93Aa |
| SMBN (mg kg$^{-1}$) | 4.80±0.77Ab | 2.06±0.36Aa | 4.07 ±0.57Aab | 3.28 ±0.80Aab | 2.34 ±0.38Aab | 5.86 ±0.73Aab | 8.31±1.65Bb | 4.16±0.89Aa | 3.10±0.99Aa | 4.36±1.38Aa |
| SMBC/SMBN | 7.84±0.90Ba | 7.79±2.72Aa | 4.79±1.49Aa | 6.44±1.84Aa | 5.24±0.62Aa | 4.77±0.41Aa | 7.49±0.61Ba | 6.75±1.57Aab | 5.31±1.44Aab | 6.03 ±0.68Aab |
| BG (U g$^{-1}$) | 5.29±0.35Aa | 11.98 ±0.36Bc | 14.26 ±0.60Ad | 19.49 ±0.60Be | 6.99±0.43Ab | 16.32 ±1.17Bb | 6.44±0.70Aa | 11.67 ±1.00Aab | 11.09 ±0.71Aab | 14.06 ±0.29Bb |
| NAG (U g$^{-1}$) | 0.89±0.14Aa | 1.48±0.14Aa | 1.21±0.22Aa | 2.64±0.16Ab | 2.79±0.27Ab | 2.30±0.32Ba | 3.50±0.83Ba | 2.73±0.32Ba | 2.63±0.18Aa | 2.85±0.84Aa |
| DHA (U g$^{-1}$) | n.d. | n.d. | n.d. | n.d. | n.d. | n.d. | n.d. | n.d. | n.d. | n.d. |

**Note:** Values in a column with different lowercase letters represent significant difference between different N addition levels under same site (One-way ANOVA followed by LSD test, $P < 0.05$); those with different capital letters represent significant difference between different site under same N addition level ($P < 0.05$). Background, Initial value measured in mid-May; n.d., not detected.

Nitrogen addition treatments are no N addition (Control), 5 g N m$^{-2}$ yr$^{-1}$ (N5), 10 g N m$^{-2}$ yr$^{-1}$ (N10), and 15 g N m$^{-2}$ yr$^{-1}$ (N15).

The results of two-way ANOVA showed that site type, N addition and their interactions had a significant effect on SMBC, and SMBN was significantly affected by site type and the interactions between site type and N addition ($P < 0.01$, Table 3).

## Changes in soil enzyme activities

N addition changed soil enzyme activities, but the effect depended on the enzyme and the sampling site (Table 2). DHA activity at both sampling sites was below the detection limit, so in the rest of this paper, we focus on changes of the BG and NAG activities. In the sandy grassland, N addition significantly increased BG activity compared with the control in N5 and N10, but significantly decreased BG activity in N15 ($P < 0.05$). All N addition significantly increased BG activity compared with the background ($P < 0.05$). NAG activity had no significantly difference in control and N5, but increased significantly in N10 and N15 ($P < 0.05$). In the semi-

**Table 3. Two-way ANOVA results of site type, N addition and their interactions on soil microbial biomass indices (soil microbial biomass carbon (SMBC), soil microbial biomass nitrogen (SMBN), and SMBC/SMBN ratio) and soil enzyme activities (β-1,4-glucosidase (BG), soil N-acetyl-β-D-glucosidase (NAG), and soil dehydrogenase activity (DHA)) in the topsoil (to a depth of 20 cm).**

| Parameter | Site type | N addition | Site type×N addition |
|---|---|---|---|
| SMBC | 20.12** | 4.83** | 5.15** |
| SMBN | 10.81** | 1.25 | 4.64** |
| SMBC/SMBN | 0.02 | 1.29 | 0.43 |
| BG | 29.91** | 41.86** | 53.64** |
| NAG | 8.36** | 2.31 | 3.00 |
| DHA | —— | —— | —— |

Note: Significant levels (ANOVA followed by LSD test): **, P < 0.01; *, P < 0.05. SMBC, soil microbial biomass carbon; SMBN, soil microbial biomass nitrogen; BG, soil β-1,4-glucosidase activity; NAG, soil β-1,4-N-acetylglucosaminidase activity; DHA, soil dehydrogenase activity.

fixed sandy land, N addition significantly increased BG activity compared with the control in all three treatments ($P < 0.05$), but there was no significant difference between N5, N10, and N15 ($P > 0.05$). NAG activity did not differ significantly among the treatments and background ($P > 0.05$). The results of two-way ANOVA showed that site type, N addition and their interactions had a significant effect on BG, and NAG was significantly affected by site type and the interactions between site type and N addition ($P < 0.01$, Table 3).

## Microbial abundance

We used $q$PCR to determine the gene copy numbers for the total bacteria and fungi species at the two sampling sites (Fig 1). For bacteria, the 16S RNA gene copy numbers in the sandy grassland and semi-fixed sandy land ranged from $7.88\times10^7 \pm 1.41\times10^7$ to $9.91\times10^7 \pm 1.09\times10^7$ copies/g and from $7.11\times10^7 \pm 0.80\times10^7$ to $20.7\times10^7 \pm 6.09\times10^7$ copies/g, respectively. There were no significant differences in soil bacterial abundance between the sandy grassland and semi-fixed sandy land ($P > 0.05$). For fungi, the ITS RNA gene copy numbers in the sandy grassland and semi-fixed sandy land ranged from $2.44\times10^6 \pm 0.47\times10^6$ to $5.19\times10^6 \pm 2.41\times10^6$ copies/g and from $1.15\times10^6 \pm 0.16\times10^6$ to $11.3\times10^6 \pm 5.52\times10^6$ copies/g, respectively. There were also no significant differences in soil fungal abundance between the sandy grassland and semi-fixed

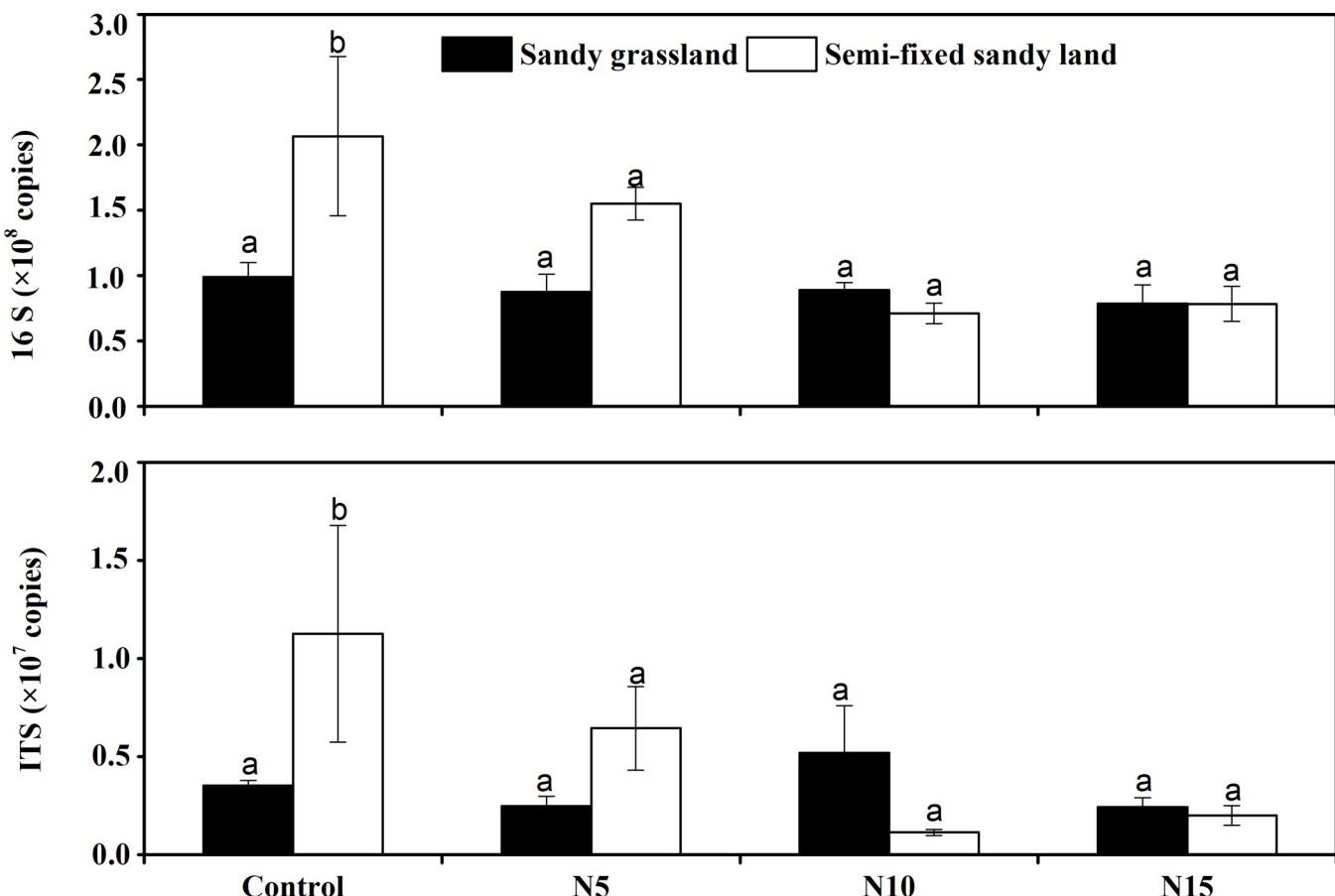

**Fig 1. Comparison of the bacterial 16S rRNA gene and fungal ITS rRNA gene copy numbers from the soils at the two sampling sites as determined by $q$PCR.** Values are means ± SD. Nitrogen addition treatments: Control, no N addition; N5, 5 g N m$^{-2}$ yr$^{-1}$; N10, 10 g N m$^{-2}$ yr$^{-1}$; N15, 15 g N m$^{-2}$ yr$^{-1}$.

sandy land ($P > 0.05$). In addition, the bacterial and fungal abundance did not differ significantly between sandy grassland and semi-fixed sandy land at any N addition level ($P > 0.05$).

## MiSeq sequencing and α-diversity indices

We obtained 1,131,376 valid reads and 2397 OTUs from the 24 samples through Illumina MiSeq sequencing analysis and classified the bacteria in these samples. Each library contained from 40,037 to 51,789 reads, with 1811 to 2086 different phylogenetic OTUs. The average length of high-quality sequences ranged from 412.779 to 417.805 bp. Similarly, we obtained 1,604,773 valid reads and 473 OTUs for fungi, and each library contained from 56,652 to 73,089 reads, with 161 to 373 different phylogenetic OTUs. The average length of high-quality sequences ranged from 230.261 to 250.127 bp.

Rarefaction curves approached saturation in all samples, indicating that the data volume in the sequenced reads was reasonable, and the discovery of a high number of reads contributed relatively little to the total number of OTUs. The curves show that only a very small fraction of the new phylotypes of the bacteria was retrieved after 50,000 sequencing reads, while the fungi was retrieved after 10,000 sequencing reads. This rarefaction curve indicated the presence of low variation in the total number of OTUs among the different samples (S1 Fig).

We estimated the α-diversity based on the observed species using the ACE, Chao, Shannon, and Simpson diversity indices. The results for bacterial and fungal diversity are summarized in S1 and S2 Tables, respectively. The observed species score (number of OTUs) for the bacterial communities ranged from 1811 to 2086, and the ACE and Chao scores ranged from 4242.927 to 5532.818 and from 4119.895 to 5489.310, respectively. The Shannon and Simpson scores ranged from 6.159 to 6.840 and 0.0030 to 0.0153, respectively. The species score (number of OTUs) for the fungal communities ranged from 161 to 373, and the ACE and Chao scores ranged from 173.659 to 395.885 and from 172.143 to 397.459, respectively. The Shannon and Simpson scores ranged from 1.386 to 4.081 and 0.039 to 0.581, respectively.

## Taxonomic composition

The samples yielded different numbers and abundance of OTUs. Sequences that could not be classified into any known group or that had an undetermined taxonomic position were assigned as unclassified or no rank group, respectively.

The bacterial OTUs were assigned into 26 phyla, 236 families, and 475 genera. Of the prokaryotic phylotypes, 10 of the 26 phyla were common to the 24 libraries: Actinobacteria, Bacteroidetes, Chloroflexi, Cyanobacteria, Firmicutes, Gemmatimonadetes, Nitrospirae, Patescibacteria, and Proteobacteria (Fig 2A), and comprised more than 98% of the total reads in every library. Proteobacteria and Actinobacteria were the two most abundant groups, comprising approximately 31.6% (357,483 reads) and 30.7% (347,423 reads) of the total reads across all samples, respectively. However, the proportions of Firmicutes and Actinobacteria varied widely among the samples, with values ranging from 19.8 to 48.1% and from 13.5 to 38.3%, respectively. The proportion of Proteobacteria reached their lowest value in sample G3N15, which was significantly different from that in samples of G1N15 and G5N15. (Sample names are defined as the site type [G, sandy grassland; S, semi-fixed sandy land] followed by the N addition treatment.) Acidobacteria and Chloroflexi were the third- and fourth-most abundant groups, comprising 14.7% (166,330) and 10.17% (115,083) of the reads, respectively, across all samples. Members of the Bacteroidetes, Gemmatimonadetes, Firmicutes, Patescibacteria, Cyanobacteria, and Nitrospirae accounted for 3.4% (38,273 reads), 3.2% (36,720 reads), 2.4% (27,384 reads), 1.3% (14,819 reads), 0.6% (6,537 reads), and 0.5% (6,046 reads) of the

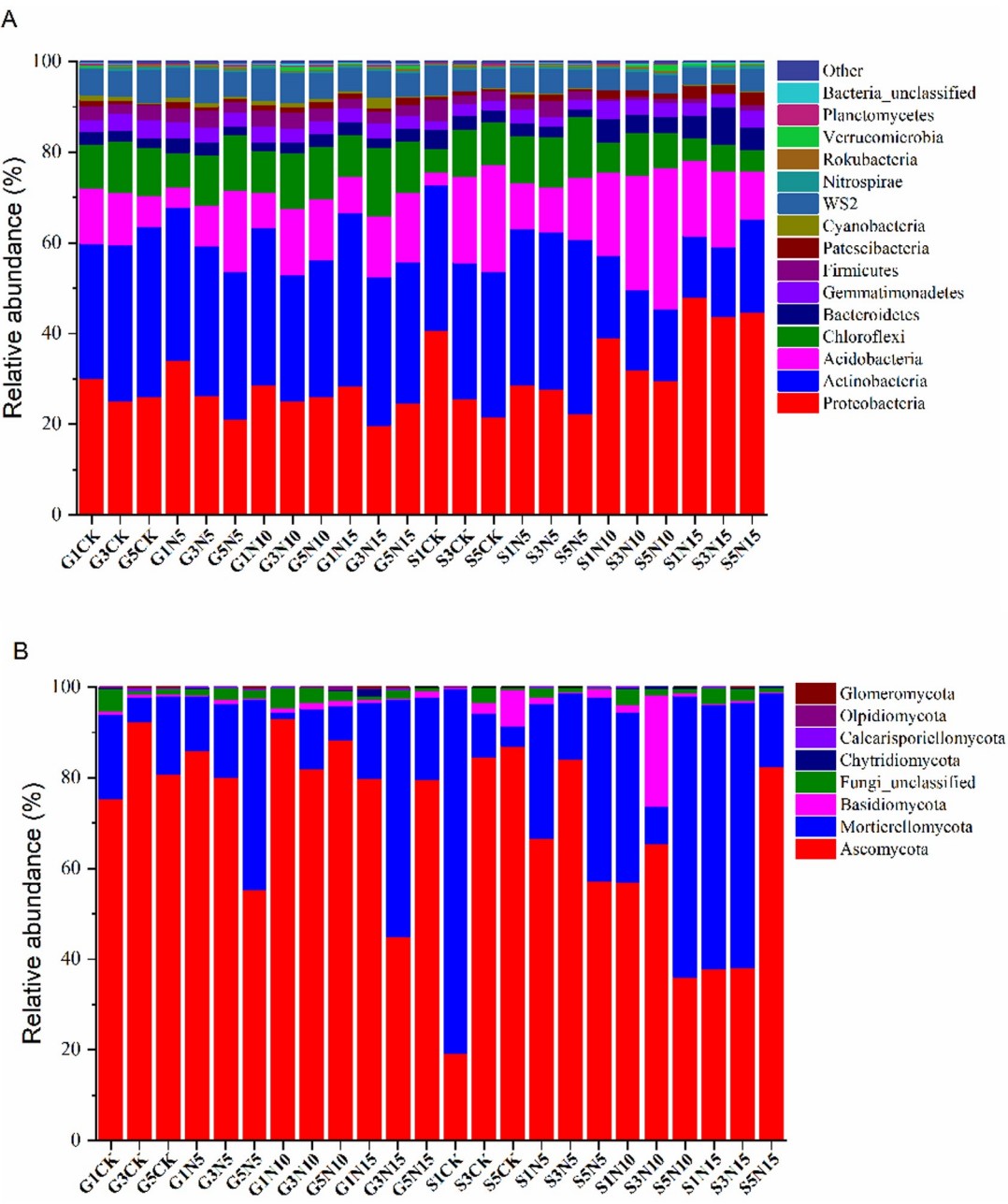

**Fig 2.** A. Relative abundance of phylotypes in the bacterial community. Sample names are the sampling location (G, sandy grassland; S, semi-fixed sandy land) followed by the nitrogen addition treatment: Control, no N addition; N5, 5 g N m$^{-2}$ yr$^{-1}$; N10, 10 g N m$^{-2}$ yr$^{-1}$; N15, 15 g N m$^{-2}$ yr$^{-1}$. B. Relative abundance of phylotypes in the fungal community. Sample names are the sampling location (G, sandy grassland; S, semi-fixed sandy land) followed by the nitrogen addition treatment: Control, no N addition; N5, 5 g N m$^{-2}$ yr$^{-1}$; N10, 10 g N m$^{-2}$ yr$^{-1}$; N15, 15 g N m$^{-2}$ yr$^{-1}$.

reads in all libraries combined. The other groups represented a small fraction (ca. 1.4%) of the total bacterial community.

The fungal communities were assigned to 8 phyla, 105 families, and 167 genera. Among them, Ascomycota was the dominant group, comprising 69.0% (1,107,374 reads) of the total reads (Fig 2B). Mortierellomycota was the second-largest group, accounting for 26.6%

(427,349 reads) of the total reads. However, the proportions of Ascomycota and Mortierello-mycota varied widely among the samples, accounting for 19.2 to 93.0% of the reads and 14.9 to 80.4% of the reads, respectively. The proportion of Ascomycota reached its lowest value in S1CK, and was significantly different from the proportions in samples of S3CK and S5CK The other fungal phyla accounted for only 4.4% of the total (70,050 reads): Basidiomycota (2.1%, 34,190 reads), Fungi_unclassified (1.9%, 30,215 reads), Chytridiomycota (0.2%, 3201 reads), Calcarisporiellomycota (0.1%, 1,383 reads), Olpidiomycota (<0.1%, 826 reads), and Glomero-mycota (<0.1%, 235 reads).

## Microbial community structure

To analyze the similarity of the bacterial communities in the different samples, we constructed a heatmap using hierarchical cluster analysis. For bacteria, the heatmap was based on the 50 most abundant bacterial genera, and this divided the bacteria into two main groups (Fig 3A). One was mainly composed of genera from the SN10 group, including S3N10 and S5N10, and grouped them with S5N15, S1N15, S1N10, and S3N15; the other grouped the members from the other samples together. The PCA results also revealed that bacterial communities from samples of SN10 group and SN15 group were grouped together at the right side of the graph

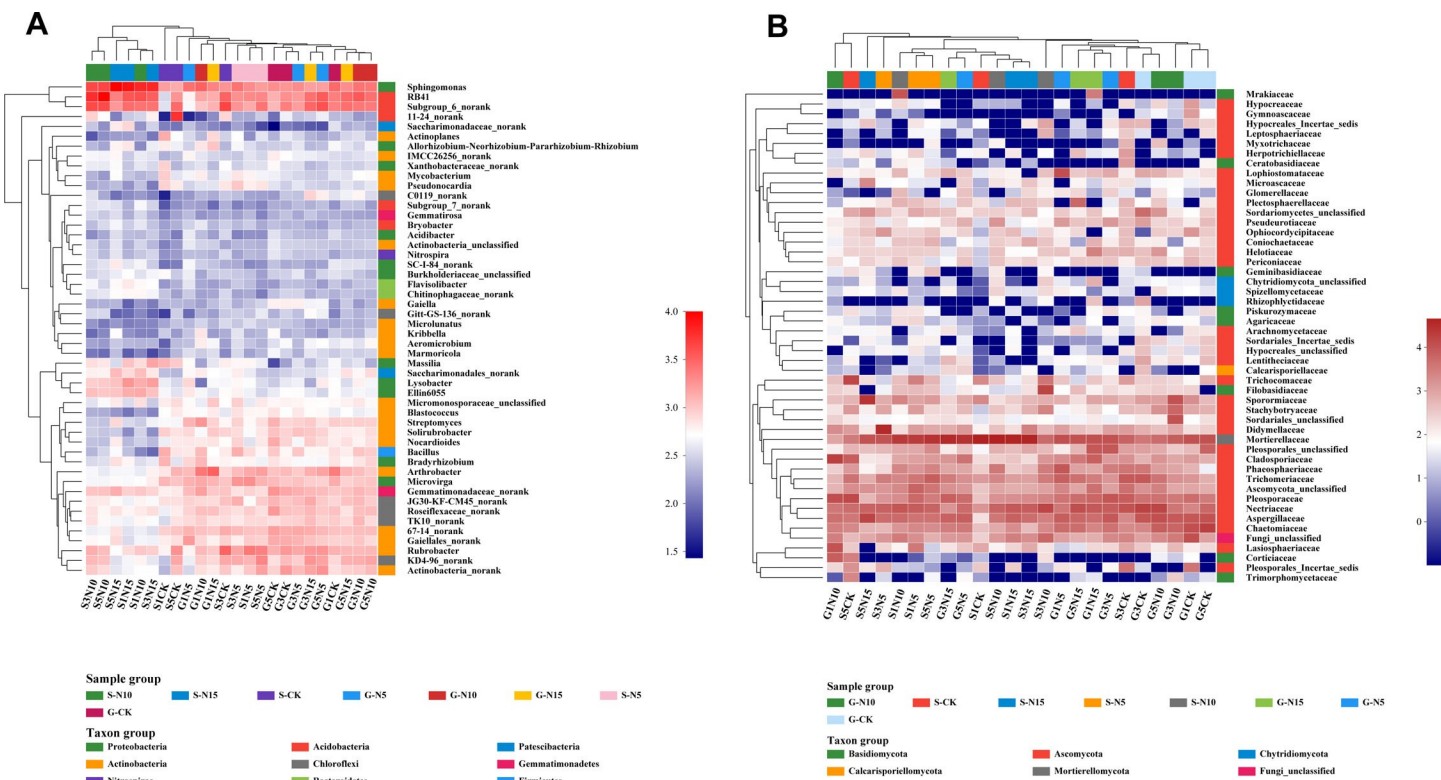

**Fig 3.** A. Heat map representations and cluster analysis for the microbial community based on 24 samples from the two sampling sites. Bacterial distributions for the 50 most-abundant genera and families. The double hierarchical dendrogram shows the bacterial and fungal distribution. Bacterial and fungal phylogenetic trees were calculated using the neighbor-joining method. Sample names are composed of the sampling site (G, sandy grassland; S, semi-fixed sandy land) and nitrogen addition: Control (CK), no N addition; N5, 5 g N m$^{-2}$ yr$^{-1}$; N10, 10 g N m$^{-2}$ yr$^{-1}$; N15, 15 g N m$^{-2}$ yr$^{-1}$. B. Heat map representations and cluster analysis for the microbial community based on 24 samples from the two sampling sites. Fungal distributions for the 50 most-abundant genera and families. The double hierarchical dendrogram shows the bacterial and fungal distribution. Bacterial and fungal phylogenetic trees were calculated using the neighbor-joining method. Sample names are composed of the sampling site (G, sandy grassland; S, semi-fixed sandy land) and nitrogen addition: Control (CK), no N addition; N5, 5 g N m$^{-2}$ yr$^{-1}$; N10, 10 g N m$^{-2}$ yr$^{-1}$; N15, 15 g N m$^{-2}$ yr$^{-1}$.

along PC1, whereas the other samples were grouped at the left along PC1, with PC1 accounting for 39.4% of the total variations (Fig 4A). PC2 only accounted for 19.8% of the variation, but again separated the samples of SN10 and SN15 group from the other samples.

For fungi, the heatmap was based on the top 50 families (Fig 3B). The samples could be divided into two clusters at the family level: one was mainly composed of the samples G1N10, S5C, S5N15, S3N5, and grouped with S1N10, S1N5, S5N5, G3N15, G5N5, S1C, S5N10, S1N15, and S3N15, and the other cluster grouped the rest of the samples together. The PCA plot grouped the fungal communities from the samples G5N5, S1N5, S3N15, G3N15, S5N10, and S1C together to the right along PC1, which accounted for 38.8% of the variation, and PC2 (which accounted for 12.5% of the variation) produced the same separation of the two sample groups (Fig 4B).

## Discussion

### The effects of N addition on soil microbial biomass indices

Our result showed the peaking growing season of SMBC and SMBN were significant higher those of background (Table 2), which in consistent with previous studies [48,49]. The SMBC and SMBN were lower in the peak growing season and higher in the dormancy season. This may be due to the large demand for soil nutrients by plants at the peak of the growing season limited the availability of nutrients by soil microbes, so SMBC and SMBN were decreased in the peaking growing [50].

Our goal was to determine the effects of changes in the soil microbial biomass indices (SMBC, SMBN, SMBC/SMBN) resulting from N addition at different levels. We found no significant effects of short-term N addition on these indices in the sandy grassland, which agrees with previous reports [51–53]. In the short term, the activity of soil microbes is regulated more strongly by plants than by the direct effects of N addition [53]. The plants competed for

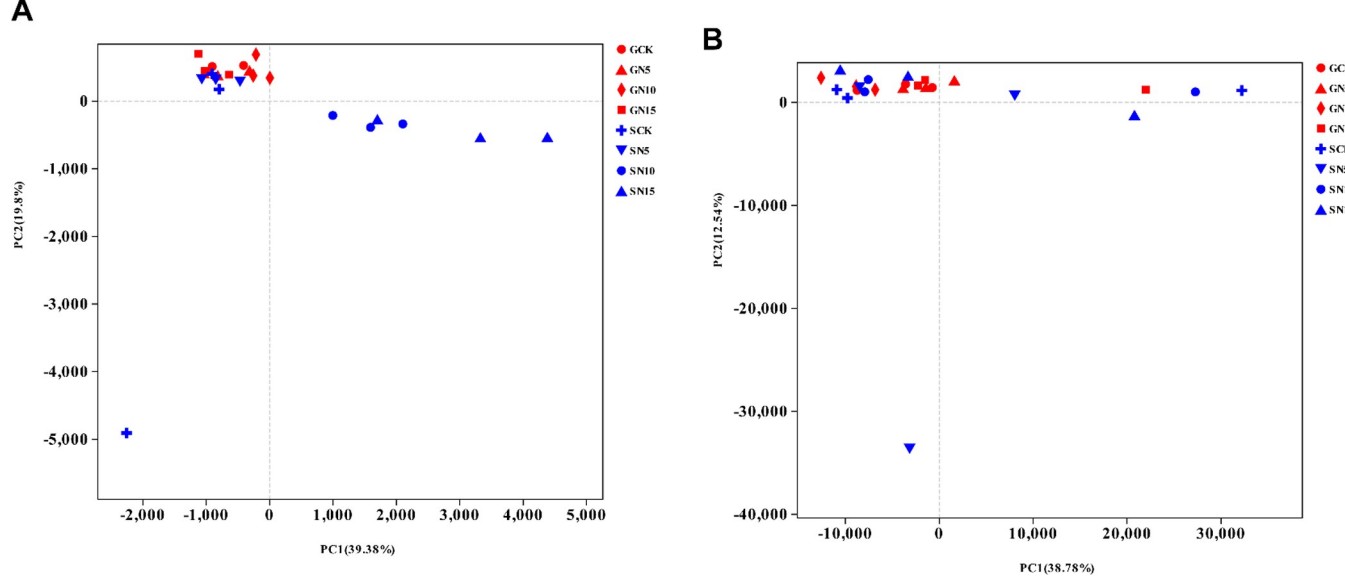

**Fig 4.** A. Results of the principal-components analysis (PCA) on bacterial communities. Plots are based on the UniFrac distance. Sample names are composed of the sampling site (G, sandy grassland; S, semi-fixed sandy land) and nitrogen addition: Control (CK), no N addition; N5, 5 g N m$^{-2}$ yr$^{-1}$; N10, 10 g N m$^{-2}$ yr$^{-1}$; N15, 15 g N m$^{-2}$ yr$^{-1}$. B. Results of the principal-components analysis (PCA) on fungal communities. Plots are based on the UniFrac distance. Sample names are composed of the sampling site (G, sandy grassland; S, semi-fixed sandy land) and nitrogen addition: Control (CK), no N addition; N5, 5 g N m$^{-2}$ yr$^{-1}$; N10, 10 g N m$^{-2}$ yr$^{-1}$; N15, 15 g N m$^{-2}$ yr$^{-1}$.

nutrient elements strongly than the microbes for the available N, and the original relationships among soil microbes may not change immediately in response to N addition. This may explain why we observed no difference in soil microbial biomass among the short-term N addition treatments in the sandy grassland. This could be attributed to the small changes in the plant community during the first growing season [53]. However, the effect of N addition on grassland soil microbial biomass depended on the amount, type, time, and initial level of nitrogen addition in grasslands in previous research. It is widely accepted that long-term N addition decreases soil microbial biomass [54–56], mainly because under natural conditions, the promoting effect of N addition on plant growth will decrease over time, resulting in a reduction of the amount of plant residues and litter that is input into the soil, and the increase of soil microbial biomass will then be inhibited by a lack of sufficient carbon [55]. We plan to continue the N addition experiments at the same sites to explore long-term changes in the microbial biomass indices.

In the semi-fixed sandy land, short-term N significantly decreased SMBC and SMBN, but their ratio was unaffected by the N addition. *Caragana microphylla* is the dominant species in the ecosystem, and the species has solid nitrogen capacity, so N addition could reduce the quantity and quality of its root exudates, thereby decreasing SMBC and SMBN [57]. Some studies have shown that the addition of N alleviated the N limitation of the soil, but that the plants decreased their allocation of resources to belowground biomass by decreasing root growth and releasing of easily decomposed materials, thereby inhibiting the growth of microorganisms [58,59]. In addition, the lack of C input to the soils represents a substrate limitation that can decrease SMBC. Dijkstra et al. (2005) [60] showed that short-term N application reduced SMBC and SMBN in tall grasses in Minnesota (United States), and explained that this resulted from a reduction of plant root secretion, resulting in an insufficient carbon source available to soil microbes, which is similar to the present results.

SMBC and SMBN in the semi-fixed sandy land were significantly higher than those in sandy grassland in the control. At the same N addition level, there was no significant difference in the microbial indices between the two sampling sites. The differences of SMBC and SMBN between the two sampling sites may have resulted from differences in the dominant plants. The nitrogen-fixing effect of the root system of *Caragana microphylla* and the existence of a large amount of root exudates may increase the soil microbial biomass compared with an annual or perennial herb-based ecosystem such as that in the sandy grassland [61].

## The effects of N addition on soil enzymes

The N5 and N10 additions significantly promoted BG activity in the sandy grassland, whereas N15 significantly decreased BG activity. N5, N10, and N15 additions significantly increased the activity of BG in the semi-fixed sandy land. Thus, N addition increased the overall activity of BG at both sampling sites. The change of BG activity reflects the variety of organic matter in the soil. With increasing N content, the N limitation for microbe decreases, and N addition promoted the accumulation and fixation of plant litter in the soil, leading to an increased carbon source for soil microorganisms to meet their demand and increasing BG activity in the soil [62]. Many studies have shown that N addition can promote BG activity [13,63,64], and our results agree with that previous research. There were no differences in BG activity between the two sampling sites for other N addition treatment, but BG activity in the control and N10 were significantly higher in the sandy grassland than in the semi-fixed sandy land. This may be because SOC and SMBC were lower in the sandy grassland than in the semi-fixed sandy land. Soil enzyme activities increase to maintain efficient utilization of soil carbon in areas with a

low soil organic carbon content [65]. This may explain why the BG enzyme activity was relatively high in the sandy grassland.

NAG is the terminal enzyme in the mineralization of soil organic nitrogen, and its degradation products can be directly used by plants and microbes. Its activity can therefore characterize soil nitrogen turnover [66]. We found that the N10 and N15 addition levels significantly increased NAG enzyme activity in the sandy grassland. The reason may be that the addition of N increased the input of plant biomass to the soil, which increased the soil organic nitrogen content and induced NAG secretion [67,68]. NAG activity in the semi-fixed sandy land did not differ among the N addition levels and at a given N addition, only the result at background, control, and N5 differed significantly between the sites, with a higher value in the semi-fixed sandy land, this may be due to the original differences between two sites. Short-term N addition did not alter overall NAG activity in the soil, and this may be because the microbial community structure did not change [69].

DHA can catalyze the redox reaction in soil, and always was used to characterize the overall activity of soil microbes [32]. Previous study had shown that long-term nitrogen addition significantly inhibits DHA activity by reducing soil pH [70]. Our result showed that the DHA was no detected, this could be attributed to low soil nutrient and low soil microbial activity in the study area, DHA activity was below the detection limit. Further research will be needed to examine DHA activity in future N addition treatment.

## The effects of N addition on the soil microflora characteristics

To compare the soil microflora characteristics of the sandy grassland and semi-fixed sandy land, we used $q$PCR and an Illumina MiSeq high-throughput sequencer to reveal differences in the microbial abundance, diversity, and community structure between the two sampling sites.

For bacteria, the number of 16S rRNA gene copies decreased compared with the control, from $20.7 \times 10^7$ to $7.8 \times 10^7$, in response to increasing N addition in the semi-fixed sandy land ($P > 0.05$), but the number of bacterial gene copies changed little after the treatment in the sandy grassland. For fungi, the dynamics of the ITS rRNA gene copies were similar to those for bacteria at both sampling sites. For example, the ITS rRNA gene copies reached the highest score, at $1.13 \times 10^7$, without N addition in the semi-fixed sandy land, and the number of copies decreased with increasing N addition. Thus, there were no significant differences in the soil bacterial and fungal abundance at either site between samples without and with N addition.

The sequencing results showed a high bacterial diversity and abundance, but a relatively low fungal diversity and abundance, at both sampling sites in the Horqin Sandy Land. For the bacteria, the Proteobacteria, Actinobacteria, and Acidobacteria were the dominant bacterial taxa. Proteobacteria and Actinobacteria are considered to be the dominant bacterial groups in many terrestrial environments, whereas Firmicutes have high resistance to high temperatures and soil moisture, and are frequently associated with arid terrestrial environments. For the fungi, the Ascomycota and Mortierellomycota were dominant. There was no obvious change of the bacterial and fungal community structures at both sites in response to the N addition treatments.

In the present study, soil microbial abundance, phylogenetic α-diversity, and the community's taxonomic structure were insensitive to the short-term N addition in the Horqin Sandy Land, which can be partly attributed to the poor soil nutrient conditions and limited moisture supply, leading to rapid uptake of the added N by the vegetation before it could become part of SMBN. This would remain true if the competition for N between plants and soil microbes was not alleviated by the N addition [70,71]. The stability of the microbial community may result

from more than the community diversity and structure; it is also likely to be linked to a range of other vegetation and soil properties, including the plant species, the abundance and size of soil aggregates and the substrate quality. The resistance and resilience (stability) of soil microbial communities are governed by soil physical and chemical structures through their effect on the microbial community composition and physiology [72,73]. In the present study, the N addition probably failed to change the soil's physical and chemical properties and microbial community structure.

Previously, many studies concentrated on the effects of long-term nitrogen addition on soil microorganisms, and researchers found that the nitrogen application had a dual effect on soil physical-chemical properties, and these changes then altered the diversity of soil microbes and their community structure. On the one hand, application of sufficient nitrogen could improve soil nutrient conditions and facilitate microbial growth and reproduction [35,74]. On the other hand, excessive nitrogen application can lead to eutrophication and acidification of soils and can inhibit fungal growth and reproduction, especially for arbuscular mycorrhizal fungi, and this can lead to changes in the overall soil microbial community. In addition, long-term nitrogen application can significantly decrease microbial diversity in grassland soils [75–78]. It will be necessary to conduct long-term nitrogen application experiments in the Horqin Sandy Land to determine how the present results will change over time.

## Conclusion

In summary, this study analyzed the responses of soil microbial characteristics to short-term N addition in the sandy grassland and semi-fixed sandy land of the Horqin sandy land. N addition significantly increased BG activity at both sites. However, short-term N addition had no significant effects on the three soil microbial indices (SMBC, SMBN, and SMBC/SMBN), on the activity of NAG, and on soil microflora characteristics (soil microbial abundance, phylogenetic α-diversity and taxonomic structure) at the three N addition levels in two sites. The N addition in the first growing season had no significant impact on the measured soil microbes, this appears to be because the short-term N addition did not alleviate the competition for N between plants and soil microbes, and because the relationships among the original soil microbes may not have changed sufficiently to affect the community structure. These findings indicate that long-term N addition will be needed to examine the responses of soil microbes when predicting future C and N cycling under global change.

## Supporting information

**S1 Fig. Rarefaction curves.** Rarefaction curves show the number of reads 97% sequence similarity level for the different samples. (A) Bacteria, (B) Fungi. Sample names include the site type (G, sandy grassland; S, semi-fixed sandy land) and nitrogen addition treatment: control (CK), no N addition; N5, 5 g N m$^{-2}$ yr$^{-1}$; N10, 10 g N m$^{-2}$ yr$^{-1}$; N15, 15 g N m$^{-2}$ yr$^{-1}$. (TIF)

**S1 Table. Phylotype coverage and diversity estimation based on the bacterial 16S rRNA gene libraries for the samples from the MiSeq sequencing analysis.** Sample names include the site type (G, sandy grassland; S, semi-fixed sandy land) and nitrogen addition treatment: control (CK), no N addition; N5, 5 g N m$^{-2}$ yr$^{-1}$; N10, 10 g N m$^{-2}$ yr$^{-1}$; N15, 15 g N m$^{-2}$ yr$^{-1}$. (DOCX)

**S2 Table. Phylotype coverage and diversity estimation based on the fungal ITS rRNA gene libraries for the samples from the MiSeq sequencing analysis.** Sample names include the site (G, sandy grassland; S, semi-fixed sandy land) and nitrogen addition treatment: control (CK),

no N addition; N5, 5 g N m$^{-2}$ yr$^{-1}$; N10, 10 g N m$^{-2}$ yr$^{-1}$; N15, 15 g N m$^{-2}$ yr$^{-1}$.
(DOCX)

**S1 Data.**
(XLSX)

## Author Contributions

**Conceptualization:** Niu Yayi, Duan Yulong, Li Yuqiang, Wang Xuyang, Chen Yun.

**Data curation:** Niu Yayi, Duan Yulong, Li Yuqiang.

**Formal analysis:** Niu Yayi, Duan Yulong.

**Funding acquisition:** Duan Yulong, Li Yuqiang.

**Investigation:** Niu Yayi, Duan Yulong, Wang Xuyang, Chen Yun, Wang Lilong.

**Methodology:** Niu Yayi, Duan Yulong, Li Yuqiang.

**Project administration:** Duan Yulong, Li Yuqiang.

**Resources:** Duan Yulong, Li Yuqiang.

**Software:** Niu Yayi, Duan Yulong.

**Supervision:** Duan Yulong, Li Yuqiang.

**Validation:** Duan Yulong, Li Yuqiang.

**Writing – original draft:** Niu Yayi, Duan Yulong.

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
