## [Decision Letter · Decision Letter 0]

12 Jan 2021

PONE-D-20-34907

Soil microbial community responses to short-term nitrogen addition in China’s Horqin
Sandy Land

PLOS ONE

Dear Dr. Yulong Duan,

Thank you for submitting your manuscript to PLOS ONE. After careful consideration, we
feel that it has merit but does not fully meet PLOS ONE’s publication criteria as it
currently stands. Therefore, we invite you to submit a revised version of the
manuscript that addresses the points raised during the review process.

The reviewer is particularly keen about the overall rationale of the study. For
instance, why did the authors choose two closely situated field sites for exploring
this research question. Furthermore, there are some results that have not been
adequately discussed (see below the comments by the reviewer).

If you would like to make changes to your financial disclosure, please include your
updated statement in your cover letter. Guidelines for resubmitting your figure
files are available below the reviewer comments at the end of this letter.

We look forward to receiving your revised manuscript.

Kind regards,

Tanvir Shahzad

Academic Editor

PLOS ONE

Journal Requirements:

2.We note that you have indicated that data from this study are available upon
request. PLOS only allows data to be available upon request if there are legal or
ethical restrictions on sharing data publicly. For more information on unacceptable
data access restrictions, please see http://journals.plos.org/plosone/s/data-availability#loc-unacceptable-data-access-restrictions.

Reviewers' comments:

Reviewer's Responses to Questions

**Comments to the Author**

1. Is the manuscript technically sound, and do the data support the conclusions?

Reviewer #1: Yes

2. Has the statistical analysis been performed
appropriately and rigorously? 

Reviewer #1: Yes

3. Have the authors made all data underlying the
findings in their manuscript fully available?

Reviewer #1: Yes

4. Is the manuscript presented in an intelligible
fashion and written in standard English?

Reviewer #1: Yes

5. Review Comments to the Author

Reviewer #1: 

The manuscript ‘Soil microbial community responses to short-term nitrogen addition in
China’s Horqin Sandy Land (PONE-D-20-34907)’ has been reviewed.

The study investigated the effect of short-term N addition on various soil microflora
characteristics at three N addition levels in the sandy grassland and semi-fixed
sandy land.

Overall, idea of the work is good however there are few key questions to be addressed
before getting the manuscript suitable for publication in PLOS ONE. The comments are
as under.

Authors need to build the rational differentiating the variable response of microbial
communities upon short term and long-term nitrogen addition – it wasn’t accordingly
discussed. Key words should be precise and most relevant, for example site or region
presence in key words may not be important. Line 44 cut ‘they’. Authors have
mentioned about the response of microbes across forest and farmland as previously
reported, assumption behind studying the short-term N addition in sandy grassland
and semi-fixed sandy land ecosystems is to be mentioned.

The activity of β- 1,4-glucosidasewas increased upon nitrogen addition (N5, N10),
could this only be related to the release of nitrogen from plant litter degradation.
Why these two particular sites with distance of 1.5 km were chosen when the
pedoclimatic condition relying on were (supposedly) the same. For fungal
amplification, 18S rRNA could be targeted? Did authors check for inhibition test to
remove the background noise before running the qPCR assays? The bacterial gene
abundances quantified through 16S RNA gene copy numbers in the sandy grassland and
semi-fixed sandy showed a high variation but were there no significant differences.
The microbial community structure through targeting ITS region could be a possible
way to know the differences in community structure across variable soils as
discussed in taxonomic composition. Community composition analyses in nicely
described but dominant phyla may be described in result section. The conclusions
should not be repetition of results rather focus the key findings.

Sincerely,

6. PLOS authors have the option to publish the peer
review history of their article (what does this mean?). If published, this will
include your full peer review and any attached files.

If you choose “no”, your identity will remain anonymous but your review may still be
made public.

**Do you want your identity to be public for this peer review?** For
information about this choice, including consent withdrawal, please see our
Privacy Policy.

Reviewer #1: No

Comments_PONE-D-20-34907.pdf
---

## [Author Response · Author response to Decision Letter 0]

11 Mar 2021

25 January 2021

Reviewer 

PLOS ONE 

RE: Submission of the revised manuscript (No. PONE-D-20-34907): Soil microbial
community responses to short-term nitrogen addition in China’s Horqin Sandy
Land.

Dear Reviewer:

Thank you very much for your assistance in the review of our manuscript. We have
revised the manuscript carefully according to your comments.

Our detailed responses to comments are presented in the remainder of this letter. All
of revisions have been highlighted in red in the revision.

1. Authors need to build the rational differentiating the variable response of
microbial communities upon short term and long-term nitrogen addition – it wasn’t
accordingly discussed.

Thank you for your advice. We have added the discussion about the variable response
of microbial communities upon short-term and long-term nitrogen addition (lines
80-90 in the revision).

2. Key words should be precise and most relevant, for example site or region presence
in key words may not be important.

We agree with your point of view that key words should be precise and most relevant.
Follow your comments, we have deleted the site presence and added the soil microbial
communities (line 33 in the revision).

3. Authors have mentioned about the response of microbes across forest and farmland
as previously reported, assumption behind studying the short-term N addition in
sandy grassland and semi-fixed sandy land ecosystems is to be mentioned.

Based on your comment, we have added the assumption in the revision (lines 95-98 in
the revision).

4. The activity of β- 1,4-glucosidase was increased upon nitrogen addition (N5, N10),
could this only be related to the release of nitrogen from plant litter
degradation.

The activity of β- 1,4-glucosidase mainly releases glucose from cellulose and plays
an important role in C cycling (lines 72-74 in the revision).The reason for the
activity of β- 1,4-glucosidase was increased upon nitrogen addition (N5, N10) has
shown in revision (lines 469-472 in the revision).

5. Why these two particular sites with distance of 1.5 km were chosen when the
pedoclimatic condition relying on were (supposedly) the same.

The region’s sandy grassland grows on aeolian sandy soils or areas with sandy soils
as the substrate and is typical of the grassland vegetation that develops in sandy
land (Munkhdalai et al., 2007), the vegetation coverage is about 60%. The dominant
plant species were annual herbs, including Artemisia scoparia, Setaria viridis,
Salsola collina, and Corispermum hyssopifolium. Semi-fixed sandy land refers to the
dune or sandy land where the vegetation coverage is between 10% and 29% and the
distribution is uniform, and the movement of wind-sand flow is blocked, but the
texture of quicksand still exists widely, the dominant plant species were perennial
shrubs Caragana microphylla and annual herbs. Therefore, we tried to explore the
response of soil microorganisms in different habitat types to N addition under the
process of desertification restoration.

Munkhdalai, Z. A., Feng, Z. W., Wang, X. K., and Sun, H. W.: Sandy grassland blowouts
in Hulunbuir, northeast China: geomorphology, distribution, and causes, Prog. Nat.
Sci.-Mater., 17, 68–73, https://doi.org/10.1080/10020070612331343227, 2007.

6. For fungal amplification, 18S rRNA could be targeted? 

Yes, 18S rRNA could be targeted for fungal amplification, meanwhile, 18S primers also
detected the Vertebrata, Ciliophora, Arthropoda and Bicosoecida (Duan et al., 2018).
In this study, we only focused on fungi amplification, therefore, we used ITS rRNA
for fungal amplification in our research.

Duan, Y. L., Wu, F. S., Wang, W. F., Gu, J. D., Li, Y. F., Feng, H. Y., Chen, T.,
Liu, G. X., An, L. Z., 2018. Differences of microbial community on the wall
paintings preserved in situ and ex situ of the Tiantishan Grottoes, China. Int.
Biodeterior. Biodegrad.

7. Did authors check for inhibition test to remove the background noise before
running the qPCR assays?

Yes, we have checked for inhibition test to make sure the accuracy of qPCR data
(lines 204 in the revision).

8. The bacterial gene abundances quantified through 16S RNA gene copy numbers in the
sandy grassland and semi-fixed sandy showed a high variation but were there no
significant differences. The microbial community structure through targeting ITS
region could be a possible way to know the differences in community structure across
variable soils as discussed in taxonomic composition.

Thank you for your advice. This comment is great helpful to our future research.
However, the bacterial gene abundances and the fungal gene abundances were
quantified through 16S RNA and ITS rRNA, respectively. The result showed that the
bacterial gene abundances and the fungal abundances in the sandy grassland and
semi-fixed sandy were no significant differences (lines 507-508 in the revision).
Therefore, ITS rRNA may not be as a possible way to explore the differences in
community structure in present study.

9. Community composition analyses in nicely described but dominant phyla may be
described in result section.

We have added the dominant phyla of the bacterial operational taxonomic units (OTUs)
and fungal communities and highlighted in red (lines 342-345, 347, 352, 364-367 in
the revision).

10. Figures could not be accessed as these were missing in the manuscript file so
editor may verify the figures quality. I evaluated the work based on the detail of
the manuscript provided. 

We separately uploaded all the figures during the manuscript submission process, if
these figures did not meet the requirements, we can re-upload all the figures to
make them accessible.

11. The conclusions should not be repetition of results rather focus the key
findings.

We have revised the description according to your comments (Line 542-553).

Thanks for your efforts to improve our manuscript. We hope that our replies and the
resulting changes will be satisfactory, but we will be happy to work with you to
resolve any remaining issues.

Sincerely,

Yulong Duan, Ph.D

Northwest Institute of Eco-Environment and Resources

Chinese Academy of Sciences

320 Donggang West Road, Lanzhou, 730000, China

Phone/Fax: 86-931-496-7219

E-mail: duanyulong@nieer.ac.cn

to Reviewers.docx
---

## [Decision Letter · Decision Letter 1]

15 Apr 2021

Soil microbial community responses to short-term nitrogen addition in China’s Horqin
Sandy Land

PONE-D-20-34907R1

Dear Dr. Yulong Duan,

We’re pleased to inform you that your manuscript has been judged scientifically
suitable for publication and will be formally accepted for publication once it meets
all outstanding technical requirements.

Kind regards,

Tanvir Shahzad

Academic Editor

PLOS ONE

Additional Editor Comments (optional):

Reviewers' comments:

Reviewer's Responses to Questions

**Comments to the Author**

1. If the authors have adequately addressed your comments raised in a previous round
of review and you feel that this manuscript is now acceptable for publication, you
may indicate that here to bypass the “Comments to the Author” section, enter your
conflict of interest statement in the “Confidential to Editor” section, and submit
your "Accept" recommendation.

Reviewer #1: All comments have been addressed

2. Is the manuscript technically sound, and do the data
support the conclusions?

Reviewer #1: Yes

3. Has the statistical analysis been performed
appropriately and rigorously? 

Reviewer #1: Yes

4. Have the authors made all data underlying the
findings in their manuscript fully available?

Reviewer #1: Yes

5. Is the manuscript presented in an intelligible
fashion and written in standard English?

Reviewer #1: Yes

6. Review Comments to the Author

Reviewer #1: The revised version of the manuscript ‘Soil microbial community
responses to short-term nitrogen addition in China’s Horqin Sandy Land
(PONE-D-20-34907R1)’ has been reviewed.

Dear Editor,

Authors have now revised the manuscript in light of comments. Thanks to authors for
taking into account my previous comments and recommendations. Currently, the article
is sufficiently improved, and is in suitable form for publication in PLOS ONE.
However, figures could not be accessed as these were missing in the manuscript file
so editor may verify the figures quality. I evaluated the work based on the detail
of the manuscript provided.

7. PLOS authors have the option to publish the peer
review history of their article (what does this mean?). If published, this will
include your full peer review and any attached files.

If you choose “no”, your identity will remain anonymous but your review may still be
made public.

**Do you want your identity to be public for this peer review?** For
information about this choice, including consent withdrawal, please see our
Privacy Policy.

Reviewer #1: No

---

## [Editor Report · Acceptance letter]

11 May 2021

PONE-D-20-34907R1 

Soil microbial community responses to short-term nitrogen addition in China’s Horqin
Sandy Land 

Dear Dr. Yulong:

I'm pleased to inform you that your manuscript has been deemed suitable for
publication in PLOS ONE. Congratulations! Your manuscript is now with our production
department. 

Kind regards, 

on behalf of

Dr. Tanvir Shahzad 

Academic Editor

PLOS ONE